# Spotlight on Nociceptin/Orphanin FQ Receptor in the Treatment of Pain

**DOI:** 10.3390/molecules27030595

**Published:** 2022-01-18

**Authors:** Amal El Daibani, Tao Che

**Affiliations:** 1Department of Anesthesiology, Washington University School of Medicine, St. Louis, MO 63110, USA; aeldaibani@wustl.edu; 2Center for Clinical Pharmacology, University of Health Sciences and Pharmacy in St. Louis, St. Louis, MO 63110, USA

**Keywords:** nociceptin/orphanin FQ receptor, NOP receptor, ligands, opioid receptor, nociceptin, N/OFQ, analgesia

## Abstract

In our society today, pain has become a main source of strain on most individuals. It is crucial to develop novel treatments against pain while focusing on decreasing their adverse effects. Throughout the extent of development for new pain therapies, the nociceptin/orphanin FQ receptor (NOP receptor) has appeared to be an encouraging focal point. Concentrating on NOP receptor to treat chronic pain with limited range of unwanted effects serves as a suitable alternative to prototypical opioid morphine that could potentially lead to life-threatening effects caused by respiratory depression in overdose, as well as generate abuse and addiction. In addition to these harmful effects, the uprising opioid epidemic is responsible for becoming one of the most disastrous public health issues in the US. In this article, the contributing molecular and cellular structure in controlling the cellular trafficking of NOP receptor and studies that support the role of NOP receptor and its ligands in pain management are reviewed.

## 1. Introduction

Persistent pain affects more than 30% of North America’s population throughout their life and it attributes to substantial expense in the US with annual costs ranging between $560 and $635 billion, which is larger than the cost of the nation’s priority health conditions [1]. This main socio-economic issue is expected to have a two-fold increase within the next 10 years especially in the elderly, as reported by the 2010 Medical Expenditure Panel Survey (MEPS). Despite the life- threatening effect caused by respiratory depression in overdose and the potential of high abuse, opioid analgesics stand as the conventional choice of treatment for moderate to severe pain [2,3,4,5,6]. As a result of misuse and extensive diversion, the use of opioids has become a leading crisis in the US, which was declared by the United States Department of Health and Human Service (HHS) in 2017 [7,8]. According to the Centers for Disease Control and Prevention (CDC), a significant increase in overdose-related deaths occurred in 2020 in which the involvement of synthetic opioids was over 60% [9]. For this reason, several research institutes have made it a priority to develop safe, effective, and non-addictive therapeutics for chronic pain management and address opioid-use disorders with innovative medications, to save lives and encourage recovery.

Opioids exert their effect via opioid receptors, a member of a large superfamily of seven-transmembrane-spanning (7TM) G-protein-coupled receptors (GPCRs), mu (MOP receptor), kappa (KOP receptor), delta (DOP receptor), and nociceptin (NOP receptor) [10]. Since NOP receptors are distributed in various regions (dorsal root ganglia (DRG), spinal dorsal horn (SDH), and brain) that are involved in pain transmission, NOP receptor ligands are under investigation primarily as an alternative for MOP receptor opioid analgesic in addition to their anxiolysis and antidepressant-like effect [11,12,13]. However, the NOP receptor was considered a controversial drug target for analgesics because of its unique pharmacological effects in pain modulation (antinociceptive vs. nociceptive effects) in the earlier phases of discovering nociceptin [14,15,16,17,18,19]. Currently, the NOP receptor has become a main focus as a promising target for analgesics as NOP receptor ligands have reported to show antinociceptive effects in non-human primates regardless of their administered doses and administration routes (spinal or supraspinal).

Moreover, the bifunctional and multifunctional NOP/opioid receptor agonists have recently displayed potent antinociceptive activity with favorable side effect profiles. Among these agonists, cebranopadol represents a promising therapeutic candidate for pain, according to the results of its clinical trials. In this article, the current literature for NOP receptor’s crystal structure, distribution, signaling pathway, and the rational design of NOP receptor ligands with various pharmacological profiles as a promising alternative for conventional opioid analgesic is reviewed to assess its therapeutic potential as analgesics.

## 2. Structure of NOP Receptor

In the mid 1990s, the human cDNA clone that encodes the NOP receptor was first isolated from the human and mouse brainstem and was then identified in several murine genomes including rat, pig, and guinea pig [20,21,22,23,24,25]. It was previously known as “orphanin FQ”, “nociceptin,”, or “ORL-1” for opioid receptor-like 1 receptor, due to the lack of its endogenous peptide ligand in the binding assays; however, nociceptin or orphanin FQ (N/OFQ) that closely resemble dynorphin A, a selective KOP receptor endogenous peptide, was characterized a year later by applying reverse pharmacology as the endogenous neuropeptide for NOP receptor [14,15]. This endogenous neuropeptide has 17 amino acids, Phe-Gly-Gly-Phe-Thr-Gly-Ala-Arg-Lys-Ser-Ala-Arg-Lys-Leu-Ala-Asn-Gln, which have quite unique features. The Phe-Gly-Gly-Phe amino terminal is noticeably comparable to the Tyr-Gly-Gly-Phe that is conserved in other classical opioid peptides [26,27]. Moreover, the number of Lys and Arg residues that are found in N/OFQ are similar to dynorphin A. Along with this resemblance, the gene structure of opioid peptide genes (preprodynorphin and preproenkephalin) and nociceptin precursor gene are also similar [27,28]. Multiple conserved amino acid residues and motifs specifically in the transmembrane helices and the intracellular loops have been identified by comparing the cDNA-derived amino acid sequence of the NOP receptor protein with that of other opioid receptors, indicating that NOP receptor belongs to GPCR Class A (rhodopsin-like) receptors, as the fourth and last characterized opioid receptor [29]. Consequently, the IUPHAR nomenclature defined this receptor and its peptide which are currently named NOP receptor and N/OFQ [30].

To date, three crystal structures of human NOP receptor have been solved with three piperidine-based antagonists (Banyn Compound-24 (C24), SB-612111, and Compound-35 (C-35)) at a resolution of 3 Å [31,32]. These crystal structures provide a perspective into the atomic details of the molecular structure of the NOP receptor and support previous homology models developed to further understand the functional mechanism of NOP receptor. In all three structures, the protonated nitrogen of the piperidine interacts with the D130^3.32^ (superscripts indicate the Ballesteros Weinstein TM helix residue numbering) residue in NOP receptor which leads to the formation of a salt bridge, implying the high affinity for these ligands. Consistent with NOP receptor crystal structure in its inactive state, the previous homology models of NOP receptor in complex with N/OFQ further support that the N-terminal amino groups of an endogenous neuropeptide agonist, N/OFQ, interact with D130^3.32^ [33,34,35]. This indicates the important role of this residue which is conserved in other canonical opioid receptors on the binding of NOP receptor ligands. Moreover, the replacement of D130^3.32^ into alanine or asparagine in the mutagenesis studies abolished N/OFQ binding, emphasizing the negative charge essentiality at this location [32].

Computer aided molecular docking studies of the selective NOP receptor agonist Ro 64-6198 into the first active state NOP receptor homology model, have also indicated signs for the mentioned NOP receptor selectivity enhancing of interactivity [34]. In this model, the amide hydrogen in Ro 64-6198 directly interacts with T305^7.39^ to form a hydrogen bond that takes place at the extracellular end of the orthosteric binding site, while the phenalenyl ring of Ro 64-6198 and the hydrophobic V279^6.51^ residue interact together inside the binding site.

Despite these studies that have highlighted the key residues and structures that are involved with ligand binding, receptor activation, and signaling, the determination of NOP receptor in its active state is required to illuminate the conformational changes in receptor’s architecture.

## 3. The Distribution and Signaling Pathway of NOP Receptor

Several techniques and animal model including in situ hybridization, immunohistochemistry, autoradiography, RT-PCR, knock-in mice with a fluorescent-tagged NOP receptor (NOP receptor-eGFP) in place of the native NOP receptor, and [^35^S]-GTPγS assay were employed to reveal the tissue distribution of NOP receptor. It is widely expressed in the human and other animal species both in the central and peripheral nervous systems [12,36]. Peripherally, the human immune cells (lymphocytic B and T-cell lines, monocytic cell lines, and circulating lymphocytes and monocytes) express NOP receptor mRNA [37]. Centrally, the NOP receptor mRNA is detected in the cortical areas, hypothalamus, mammillary bodies, the substantia nigra, thalamus nuclei, limbic structures (the hippocampus and amygdala), brainstem (colliculi, the ventral tegmental area, the locus coeruleus), and the pituitary gland [37,38].

Because NOP receptor activation modulates several physiological functions and pharmacological roles including, but not limited to, pain sensation, mood, learning, memory, cardiovascular control, and immune response [39], it is important to understand its signaling pathways and subsequent trafficking events. NOP receptor has shown a high sequence identity and homology in the TM helices and intracellular loops with other classical opioid receptors (DOP receptor, MOP receptor, and KOP receptor) which couple to inhibitory G proteins, suggesting a similar activation mechanism upon ligand binding. This binding triggers the heterotrimeric dissociation of Gαβγ subunits following the replacement of guanosine diphosphate (GDP) by guanosine triphosphate (GTP) at Gα subunit and subsequently induces multiple intracellular signaling pathways [40,41]. The dissociated Gα subunit suppresses adenyl cyclase and cAMP production, while Gβγ subunits directly couple with different ion channels such as Ca^2+^ and Kir3 [42,43,44]. NOP receptor also regulates the voltage-dependent Ca^2+^ channels by modulating Rho-associated coiled-coil-containing protein kinase (ROCK) and LIM domain kinase (LIMK) [45]. Like canonical opioid receptor, the suppression of pre and postsynaptic Ca^2+^ influx, the activation of G protein gated inward rectifying potassium (GIRKs) conductance, as well as the inhibition of various ions channels such as Na^+^ channel resulted in cellular hyperpolarization and attenuation of neuronal excitability and nociceptive stimuli transmission, thus producing antinociceptive effects [46]. In addition to ion channels, the activation of NOP receptors also modulates all three mitogen-activated protein kinase (MAPK). MAPK activity thereby regulates cell proliferation, progression, and differentiation (ERK1/ERK2), as well as the response to cellular stressors (p38 and JNK1/JNK2/JNK3) [47,48]. Moreover, the neurotransmitter release of serotonin, noradrenaline and glutamate, as well as the phospholipase (PLC) A2 and C signaling, are induced by NOP receptor activation [49,50,51,52].

Within minutes of NOP receptor activation, the uncoupling of NOP receptor to G protein is facilitated by a desensitization process, a feedback mechanism to control the receptor overstimulation during acute and chronic exposure to the ligand [53]. This process is regulated by various kinases such as GPCR kinases (GRKs) that mediate the phosphorylation, and the arrestin ligation to the C-terminus of the opioid receptor. Besides GRKs, second messenger-dependent protein kinases including protein kinase A (PKA), protein kinase C (PKC), and calcium/calmodulin-dependent protein kinase II have also been shown to phosphorylate and desensitize the NOP receptor [54,55]. The receptor desensitization is suppressed through the inhibition of mitogen activated protein dependent kinase that could interfere with the arrestin recruitment. After the GRK/arrestin recruitment, the NOP receptor is translocated into the intracellular compartment through clathrin-mediated endocytosis into which the receptor is recycled and re-sensitized to restore the receptor function back again.

## 4. Ligands of NOP Receptor

Analgesia is one of the potential clinical indications of NOP receptor due to its wide distribution in the nervous system (central and peripheral) which are involved in the pain processing pathways. In this review, NOP receptor ligands including N/OFQ-related peptides, nonpeptidic, and bifunctional compounds with different pharmacological profiles (full agonist, partial agonist, and antagonist) that represent viable drug target for pain are spotlighted. Initially, intrathecally (i.t) administered N/OFQ produces dose-dependent analgesia in the tail flick assay and flinching behavior in the formalin test without causing sedation as well as promotes antinociceptive effect of morphine in both rats and monkey [16,17,18,19]. Whereas opposite effects like hyperalgesia and a decrease in locomotor activity are produced as a result of the intracerebroventricular (i.c.v) N/OFQ administration in the hot plate test and the tail flick test in mice [14,15]. The unexpected action of i.c.v. N/OFQ administration resulted from both the anti-opioid activity (antagonizing MOP receptor, DOP receptor, and KOP receptor antinociception activity) via NOP receptor stimulation in the periaqueductal gray (PAG) neurons and the reversal of opioid induced analgesia of N/OFQ opposed to the nociceptive activity as proposed previously [56,57,58]. These findings indicate the dual actions of N/OFQ that mainly depend on the administered dose, pain models (chronic or acute), examined species, and the route of administration as illustrated in Figure 1. The reason behind this discrepancy across species is not known yet; however, some studies (reviewed in [29,59]) suggest that the difference in neuronal circuitry of pain between different species could be the reason for NOP receptor system having opposite effects in pain processing. Furthermore, the effectiveness of NOP receptor agonists in addressing chronic pain (over acute pain), can be explained by the varying levels of NOP receptor mRNA and NFQ peptide induced by chronic inflammation.

In this section, the relevant pharmacological features of NOP receptor ligands including peptide, nonpeptide, and bifunctional and mixed NOP receptor compounds are explored with a focus on their role in modulating pain to further comprehend the nature of the N/OFQ–NOP receptor system within these processes.

### 4.1. Peptide Ligands Related to N/OFQ Targeting Pain

Earlier systematic SAR studies revealed that both truncation and amidation of N/OFQ are required to sustain the biological activity of N/OFQ and avoid the N-terminus degradation by proteases, respectively [60]. As a result, N/OFQ(1-13)-NH2, which is the shortest peptide sequence that maintains the potency, efficacy, and affinity as N/OFQ, has been used as a template to design a new series of N/OFQ analogues. In the frame of SAR studies, several peptides with distinct pharmacological activity have been identified such as [Phe1Ψ(CH2-NH)Gly2]N/OFQ(1-13)NH2, UFP-101, and [Nphe1]N/OFQ(1-13)NH2 (NOP receptor antagonists), UFP-112 (highly potent NOP receptor agonist), and UFP-113 (partial NOP receptor agonist) [61,62,63,64,65]. The peptides that have antinociceptive activity are summarized in Table 1 and described below.

#### 4.1.1. [Nphe^1^]N/OFQ(1-13)NH2

A preliminary hypothesis regarding the behavior of NOP receptor-active compounds stated that if N/OFQ induces pain, antagonists are likely to exhibit antinociceptive activity. To test this, [Nphe^1^]N/OFQ(1-13)-NH_2_, the first reported peptide with antagonist activity, was generated by shifting the side chain of Phe^1^ from C to N atom in the amidated N/OFQ. A binding assay using Chinese hamster ovary (CHO) cells that express recombinant human NOP receptor and cyclic AMP accumulation in CHO cells identified the antagonistic properties of [Nphe^1^]N/OFQ(1-13)-NH_2_. The mouse tail withdrawal assay revealed that a single i.c.v. administration of [Nphe^1^]N/OFQ(1-13)NH2 increased the tail withdrawal latency time, while a combinational administration of [Nphe^1^]N/OFQ(1-13)NH2 with N/OFQ and morphine inverted the reduction of tail withdrawal latency time latency and promoted the antinociceptive effect of morphine, respectively, implying the analgesic action of this ligand [61,72].

#### 4.1.2. [Nphe^1^, Arg^14^, Lys^15^]N/OFQ-NH_2_ (UFP-101)

Previous studies have shown that the binding of C-terminus-amidated N/OFQ to the acidic restudies at the ECL2 of NOP receptor was enhanced by inserting Arg and Lys [35,73]. Combination of [Nphe^1^]N/OFQ(1-13)-NH_2_ and [Arg^14^, Lys^15^]N/OFQ-NH_2_ led to the generation of a new peptide [Nphe^1^, Arg^14^, Lys^15^]N/OFQ-NH_2_, also called UFP-101 [62]. In vitro assays including functional binding assays using (CHO cells expressing human NOP receptor and [^35^S]-GTPγS), cyclic AMP accumulation experiment, and Schild regression analysis of electrically stimulated isolated peripheral (rats, mice, and guinea pigs) and central tissues (rat) showed that UFP-101 competitively antagonized the effects of N/OFQ. Similar to [Nphe^1^]N/OFQ(1-13)-NH_2_, i.c.v. administration of 10 nmol UFP-101 produced antinociceptive effect in the mouse tail withdrawal assay, but with higher potency and longer duration of action, indicating that the presence of Arg^14^ and Lys^15^ may promote either the binding of UFP-101 to NOP receptor and/or the UFP-101 metabolic stability. Since UFP-101 is a selective NOP receptor antagonist, it has been also used as a research tool to confirm that NOP receptor mediates both the inhibition of spinal excitatory transmission in vitro as well as the spinal antinociception in vivo [66].

#### 4.1.3. [(pF)Phe^4^Aib^7^Arg^14^Lys^15^]N/OFQ-NH_2_ (UFP-112)

The chemical modifications of the phenyl ring in Phe4 residue that is essential for NOP receptor activation by inserting pF along with the replacement of Ala at position 7 by α-aminoisobutyric acid (Aib) in N/OFQ sequence resulted in generation of more potent ligands [74,75,76]. By applying these two chemical modifications to [Arg^14^, Lys^15^]N/OFQ-NH_2_, [(pF)Phe^4^Aib^7^Arg^14^Lys^15^]N/OFQ-NH_2_, also known as (UFP-112), was synthesized [67]. This ligand acts as a potent (100-fold higher than N/OFQ) and a selective NOP receptor agonist. A long-lasting dose dependent antinociceptive effect was observed after the i.t. administration of UFP-112 (1–100 pmol) in the mouse tail withdrawal assay. In contrast, the same dose of UFP-112 produced a pronociceptive effect and a long-lasting reduction in the locomotor activity when it was administered intracerebroventricularly. Subsequent to intravenous (i.v.) administration of UFP-112 in rats, diuresis as well as reduction in heart rate, blood pressure, and urinary sodium excretion were significantly observed. Consistent with the mouse tail withdrawal assay finding, a long-lasting dose dependent antinociceptive effect was also observed after the i.t. administration of UFP-112 (1–10 nmol) in monkeys without inducing itching by using acute and chronic primate pain modalities (acute noxious stimulus and capsaicin-induced thermal hyperalgesia, respectively) [68]. Notably, the spinal administration of a subthreshold dose of UFP-112 (1 nmol) synergized a morphine analgesic effect without increasing pruritus.

#### 4.1.4. [Phe^1Ψ^(CH_2_-NH)Gly^2^(pF)Phe^4^Aib^7^Arg^14^Lys^15^]N/OFQ-NH_2_ (UFP-113)

The combination of [Phe^1Ψ^(CH_2_-NH)Gly^2^](N/OFQ-NH_2_ that was synthesized to further avoid the protease degradation [63] and the mentioned above [(pF)Phe^4^Aib^7^Arg^14^Lys^15^]N/OFQ-NH_2,_ UFP-112, led to the generation of [Phe^1Ψ^(CH_2_-NH)Gly^2^(pF)Phe^4^Aib^7^Arg^14^Lys^15^]N/OFQ-NH_2_, also referred to as UFP-113 [77]. In vitro pharmacological characterization studies that include the functional [^35^S]-GTPγS binding in CHO cells that express the human NOP receptor and electrically stimulated mouse and rat vas deferens and guinea pig ileum tissues, reveals that UFP-113 acts as a selective partial agonist for NOP receptor [77]. The spinal catheterization of UFP-113 induced an analgesic response in rats at doses that range between (0.001 and 1 nmol); however, in the knockout of rats for the NOP receptor gene the analgesic effect no longer persisted, implying that the antinociceptive effect of UFP-113 is mediated through the NOP receptor stimulation [69].

#### 4.1.5. PWT2-N/OFQ

By employing a novel chemical strategy using peptide wilding approach (PWT), three tetrabranched derivatives of N/OFQ that include PWT1-N/OFQ, PWT2-N/OFQ, and PWT3-N/OFQ were generated [78]. Both in vitro ([^35^S]-GTPγS binding, calcium mobilization, and electrically stimulated mouse vas deferens assays) and in vivo studies using NOP receptor gene knocked out [NOP receptor (−/−)], revealing that these PWT derivatives act as full NOP receptor agonists that have high potency and a long duration of action of, particularly in PWT2-N/OFQ (40-fold more potent than N/OFQ) [70]. Additionally, analgesic effects were reported after the spinal administration of PWT2-N/OFQ using the nociceptive pain model (tail withdrawal assay) and the neuropathic pain model (chronic constriction injury) in mice and monkeys [71]. PWT2-N/OFQ exhibited higher potency (40-fold more potent) and longer duration (10-fold longer duration of action) in comparison to N/OFQ.

Despite having high potency and selectivity of the previously mentioned NOP receptor peptides in targeting NOP receptor, their pharmacokinetic properties, specifically their poor penetration across the blood-brain barrier have limited their therapeutic indications. However, these peptides have substantially contributed to the detailed understanding of the various responses of the peripheral (respiratory, gastrointestinal, genitourinary, immune, and cardiovascular systems) and central (pain transmission, anxiety, food intake, locomotion, and drug addiction) systems that are related to the N/OFQ–NOP receptor system.

### 4.2. Non-Peptide NOP Receptor Ligands Targeting Pain

To overcome the poor metabolic stability of peptide ligands related to N/OFQ and require to be administered either intrathecally or intracerebroventricularly, several studies were conducted to identify new selective non-peptide ligands that are suitable for intraperitoneal or oral administration. High-throughput screening and medicinal chemistry research have led to the discovery of multiple classes of chemical compounds including piperidines, spiropiperidines, nortropanes, 4-amino-quinolines, and quinazolines that act as NOP receptor ligands with enhanced metabolic stability. The non-peptides that have antinociceptive activity are summarized in Table 2 and described below.

#### 4.2.1. Ro 65-6570

The high-throughput screening of 8-acenaphthene-l-yl-l-phenyl-l,3,8-triaza-spiro[4.5]decan-4-one was performed to develop Ro 65-6570, 8-(1,2-dihydroacenaphthylen-1-yl)-1-phenyl-1,3,8-triazaspiro[4,5]decan-4-one, by a group of scientists at Roche laboratories [86]. In vitro studies that include radioligand binding and cAMP inhibition assays in (CHO) cells expressing the recombinant human NOP receptor indicated that Ro 65-6570 acts as a NOP receptor full agonist with poor selectivity in comparison to other opioid receptors [87]. In mice, i.v. administration of Ro 65-6570 resulted in dose-dependent antinociceptive effects without modifying motor coordination using formalin paw and orofacial formalin (OFF) tests [79,80]. Further in vitro functional selectivity studies such as the BRET-based assay revealed that Ro 65-6570 is a G protein-biased agonist which exhibited antinociceptive effects in β-arrestin 2 knockout mice as compared to the wild-type [88,89].

#### 4.2.2. Ro 64-6198

In an effort to develop a new NOP receptor agonist with high selectivity (greater than 100-fold over canonical opioid receptors) and potency, [(1*S*,3a*S*)-8-(2,3,3a,4,5, 6- hexahydro-1*H*-phenalen-1-yl)-1-phenyl-1,3,8-triaza- spiro[4.5] decan-4-one], also known as Ro 64-6198, was identified by a group of scientists at Hoffman La Roche in Switzerland [81,90,91]. Using Ro 64-6198 as a valuable pharmacological tool highlighted therapeutic applications for NOP receptor agonist such as anxiety, neuropathic pain, addiction, cough, and anorexia, in addition to the undesirable effects it has on learning, memory, motor activity, and body temperature (hypothermia) [92]. Similar to morphine, analgesic effects in the hot plate and shock threshold assays were observed after the systemic administration of Ro 64-6198 (3 mg/kg, intraperitoneal (i.p)) in wild-type mice but not in NOP receptor knockout mice [82,83]. Conversely, increased pain sensitivity was observed as an opposite effect in the tail flick assay, implying the complex role of NOP receptor in pain processing. Furthermore, coadministration of low doses (1 mg/kg) of Ro 64-6198 and morphine resulted in an additive analgesic effect [83]. Consistent with these findings, analgesic effects without causing depression, itching, and reinforcing responses were observed after the subcutaneous (s.c.) administration of Ro 64-6198 (0.001–0.06 mg/kg) in both acute (acute noxious stimulus) and chronic (capsaicin-induced neuropathic pain) pain modalities in monkeys [93]. Pretreatment with J-113397 (0.1 mg/kg), a selective nonpeptidic NOP receptor antagonist, blocked Ro 64-6198-induced antinociception, emphasizing that the antinociceptive actions of Ro 64-6198 is mediated via NOP receptor. Despite the robust analgesic effects of systemically administered Ro 64-6198 in non-human primates, several in vivo studies using tail flick and immersion, tactile or cold water stimulation and foot shock assays revealed that Ro 64-6198 does not modulate pain processing in rodents, except mouse hot plate assay [81,83,93,94,95].

#### 4.2.3. SCH221510

SCH221510 is a potent and selective non-peptide NOP receptor agonist that was reported to induce analgesia in neuropathic pain when administered orally and intrathecally in mice and rat models, respectively [96,97,98]. It is also reported to attenuate the respiratory depression and itch response that were observed after the systemic administration of buprenorphine to a non-human primate, as well as reinforcing MOP receptor agonists induced responses in rats [97,99]. Conversely, a s.c. administration of SCH221510 (3 and 10 mg/kg) in hot-plate test did not produce analgesia, while SCH221510 administration (3 mg/kg) reduced morphine-induced analgesia. The co-administration of SCH221510 (3 mg/kg) and morphine (10 mg/kg) accelerated the tolerance development to the antinociceptive effect of morphine in female mice [100].

### 4.3. Bifunctional and Mixed NOP Receptor Compounds

Considering the potential ability of intracerebroventricularly administered N/OFQ to attenuate morphine tolerance and suppress drug reinforcing response, the development of new synthetic agonists may constitute an innovative pharmacological approach for analgesics that target both MOP receptor and NOP receptor to enhance their analgesic effect and minimize their side effects as depicted in Figure 2 [99,101,102,103,104,105]. Additionally, multiple pathophysiological pathways are involved in the pain process, so developing analgesic agents with multiple mechanisms of actions could be an innovative strategy for developing new effective and safe analgesics [106]. Accordingly, several compounds including AT-121 (a partial agonist of NOP receptor and MOP receptor), buprenorphine (semisynthetic multifunctional opioid), and its analogue BU08028 were synthesized (reviewed in [107,108,109]).

#### 4.3.1. SR 16435

SR 16435 (Figure 3), also referred to as [1-(1-(bicyclo[3.3.1]nonan-9-yl)piperidin-4-yl)indolin-2-one] behaved as a bi-functional NOP receptor /MOP receptor partial agonist with high binding affinity was synthesized by Toll group [110]. In mice, SR 16435 administration produced an analgesic effect (s.c. and i.t.) which was effective and potent in attenuating both neuropathic and inflammatory pain (i.t) with diminished tolerance development to the antinociceptive effect of SR 16435 [96,110]. Nonetheless, the conditioned place preference (CPP) that was primarily mediated by MOP receptor activation was induced after the administration of SR 16435. This finding emphasizes that full agonistic activity at NOP receptor could be required to reduce the rewarding properties associated with MOP receptor [110].

#### 4.3.2. AT-121

AT-121 (Figure 3) is a non- morphinan compound which acts as a bifunctional NOP receptor /MOP receptor partial agonist with high binding affinity [114]. It was synthesized to optimize the pharmacological profile of MOP receptor agonists by synergizing their therapeutic effects (analgesia and treatment of substance abuse) and minimizing their side effects (respiratory depression, tolerance dependent, and abuse liability) via targeting NOP receptor. In monkeys, s.c. administration of AT-121 produced morphine-like analgesic and antiallodynic effects using the warm water tail-withdrawal assay and capsaicin-induced allodynia, respectively, without trigging itch, physical dependence, respiratory depression, and hyperalgesia mediated by opioid. These effects were confirmed to be mediated by MOP receptor and NOP receptor activation by using selective dose of MOP receptor and NOP receptor antagonists, J-113397 (0.1 mg/kg) and naltrexone (0.03 mg/kg), respectively. Additionally, AT-121 could be therapeutically implicated for opioid addiction as it lacks the abuse potential (reinforcing effects) and diminished oxycodone reinforcing response.

#### 4.3.3. Buprenorphine and Its Analog BU08028

Buprenorphine (Figure 3) is a natural derived alkaloid of the opium poppy with a mixed pharmacological activity (MOP receptor /NOP receptor partial agonist and DOP receptor /KOP receptor low partial agonist) clinically approved to treat pain and substance abuse [99,115]. In rodent, full analgesic effects were produced after the administration of buprenorphine in both chronic and acute pain models [120]. After a systemic administration of 0.01–0.1 mg/kg to a non-human primate, an antinociceptive effect was present in a dose-dependent manner. A resultant respiratory depression and itch were observed and subsequently confirmed to be induced by MOP receptor activation. These side effects associated with buprenorphine were found to be attenuated by the co-administration of an NOP receptor selective agonist such as Ro 64-6198 and SCH 221510 [99]. As such, the combination emphasized the therapeutic potential of mixed MOP receptor /NOP receptor agonists as innovative analgesics. A buprenorphine analog that is known as BU08028 (Figure 3) demonstrated a similar binding profile to buprenorphine with improved binding affinity and efficacy to NOP receptor. In mice, an intrathecal administration of BU08028 produced an analgesic effect, which was more potent than morphine in attenuating both neuropathic and inflammatory pain [96]. Consistent with these results, a systemic administration of BU08028 to a non-human primate produced a long-lasting analgesic effect (>24 h) with a reduced reinforcing effect as compared to cocaine, remifentanil, or buprenorphine and without causing respiratory depression and CVS adverse effects [121].

#### 4.3.4. BPR1M97

By applying a high-throughput screening, BPR1M97 (Figure 3) was identified as a dual agonist that produced a significant analgesic effect in a tail-flick assay in mice [122]. Both in vitro assays (radioligand binding, c-AMP production, membrane potential, β-Arrestin-2 recruitment, and internalization assays) and in vivo behavior assays (tail flick and clip, respiratory and cardiovascular functional, acetone drop, von Frey hair, charcoal meal, glass bead, locomotor activity, conditioned place preference (CPP) and naloxone precipitation assays) proved that BPR1M97 behaved as a dual agonist for MOP receptor (full agonist) and NOP receptor (G-protein biased agonist) [118]. Notably, rapid analgesic actions (more potent than morphine in cancer-induced sensory allodynia) were observed after the BPR1M97 s.c. administration with less undesirable side effect as compared to morphine.

#### 4.3.5. BU10038

A naltrexone-derived bifunctional MOP receptor /NOP receptor agonist, also referred to as BU10038 (Figure 3), behaved as a partial MOP receptor and NOP receptor agonist was synthesized by Husbands and Ko groups [117]. In non-human primate, both systemic (0.001–0.01 mg/kg) and intrathecal (3 mg) administrations of BU10038 resulted in a long-lasting antinociceptive with neither reinforcing effects nor other effects like itching, respiratory depression, and tolerance when administered repeatedly.

#### 4.3.6. JTC-801

JTC-801 (Figure 3), also referred to as [N-(4-amino-2-methylquinolin-6-yl)-2-(4-ethylphenoxymethyl) benzamide hydrochloride], behaved as a NOP receptor antagonist and was developed by a group of scientists at the Central Pharmaceutical Research Institute [119]. JTC-801 produced antinociceptive effects in a hot plate test and a formalin test using mice and rats, respectively. Although the injectable and oral formulations of JTC-801 entered Phase II of its clinical trials in both Japan and the UK to treat the neuropathic and postoperative pain, it was suspended for unknown reasons [123].

#### 4.3.7. Cebranopadol

The rational optimization strategy of spiro[cyclohexanedihydropyrano[3,4-b]indole]-amine resulted in the discovery of cebranopadol (Figure 3) that represents the first in its class to be a highly potent and efficacious antinociceptive agent with combined agonistic activity at MOP receptor, NOP receptor (subnanomolar affinity), KOP receptor, and DOP receptor (low nanomolar affinity) [111,112,124]. Behavior in vivo studies including acute and chronic pain models in rodents (tail-flick, formalin test, rheumatoid arthritis, bone cancer, spinal nerve ligation, diabetic neuropathy) further indicated the high potency and extremely long-lasting analgesic effect of cebranopadol in comparison with selective MOP receptor agonist, particularly in the chronic pain model [111,125]. Extensive preclinical safety and tolerability studies have been conducted on rodent models to reveal the possible side effects on the CNS, the respiratory system, and the gastrointestinal system (reviewed in [126]). Limited range of unwanted effects were also observed, as cebranopadol did not decrease respiratory rate, develop a tolerance, or impair the motor coordination, unlike the effects of morphine. The G-protein-biased agonistic activity at NOP receptor could be the reason behind these favorable side effect profiles of cebranopadol [125]. Notably, cebranopadol is equipotent and equi-efficacious toward the G protein activation at both MOP receptor and NOP receptor without inducing phosphorylation or NOP receptor internalization and without recruiting B-arrestin2 at NOP receptor only in BRET assay [125,127]. The noncompartmental analysis in phase I and phase II clinical trials was used to assess the pharmacokinetics profiles of cebranopadol. The maximum plasma concentration [Cmax] (4–6 h) with a long half-value duration (14–15 h) was reached after oral administration of immediate release formulation of cebranopadol. After the administration of multiple once-daily oral doses of cebranopadol in patients, the steady state was reached in nearly 2 weeks. Following single- and multiple-doses administration of cebranopadol in healthy subjects and patients, a two-compartment disposition model with first-order elimination process and a two lagged transition compartments was observed [128]. Several phase II clinical trials were conducted and listed as complete in patients suffering from acute (bunionectomy trial) and chronic (diabetic neuropathy, osteoarthritis, chronic low back pain, and diabetic polyneuropathy) pain to evaluate the efficacy, safety, and tolerability of a single oral dose of cebranopadol [129,130,131,132,133,134,135]. While most phase III clinical trials have recently proven the effectiveness, safety, and tolerability of cebranopadol when administered orally (200–1000 µg per day) to cancer patients who suffer from moderate to severe chronic pain [136,137].

## 5. Future Directions and Conclusions

In this review, the rational design of NOP receptor ligands with various pharmacological profiles as a promising alternative for conventional opioid analgesic is discussed. The crystal structure, distribution, and signaling pathway of NOP receptor are also highlighted. It is important to note that other therapeutic indications for NOP receptor in the treatment of various neurological disorders and alcohol abuse have not been explored in this review. Notably, NOP receptor-related peptides have substantially attributed in expanding our knowledge regarding the various peripheral and central responses related to N/OFQ–NOP receptor system, but their poor bioavailability has limited their therapeutic implications. Regardless of the controversial results between the spinal and supraspinal administration of endogenous neuropeptide of NOP receptor that remains poorly understood, the NOP receptor ligands exhibit favorable pharmacological activity and side effects, particularly the mixed which target multiple opioid receptors. So far, cebranopadol represents the most promising NOP receptor ligand to treat acute and chronic pain without reducing respiratory rate, developing a tolerance, or impairing the motor coordination as compared to the clinically approved opioid analgesic. However, further work needs to be done to resolve the high-resolution structure of NOP receptor in its active state to elucidate the distinct residues responsible for NOP receptor agonist binding [138]. Conceivably, a deep understanding of the NOP receptor signaling pathway and structure along with computer-aided molecular docking and behavior studies will facilitate the discovery of polypharmacological ligands that target multiple receptors including NOP receptor as new effective and safe analgesics.

## Figures and Tables

**Figure 1 molecules-27-00595-f001:**
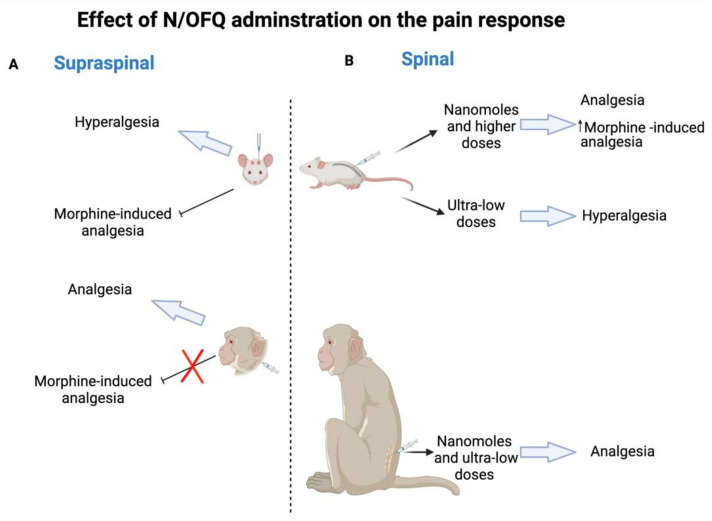
N/OFQ effect in rodent and non-human primates on pain response. (**A**) Supraspinal administration of N/OFQ produces hyperalgesia and blocks morphine-induced analgesia in rodent, whereas the opposite effect of analgesia and the promotion of an antinociceptive effect are produced in non-human primates. (**B**) Spinal administration of N/OFQ produces dose-dependent analgesia in both rodent (nanomoles and higher doses) and non-human primates (nanomoles and ultra-low doses) as well as promotes an antinociceptive effect of morphine, while ultra-low doses of N/OFQ induce hyperalgesia in rodent.

**Figure 2 molecules-27-00595-f002:**
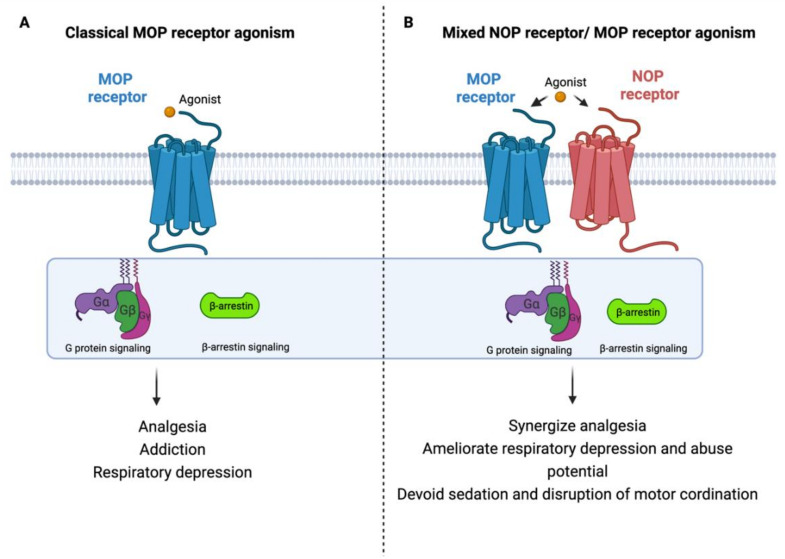
Rational design of new safer analgesics. (**A**) Beneficiary and side effects produced by MOP receptor activation. (**B**) Beneficiary (synergizing analgesic effect) and protective (ameliorating typical-opioid side effect profile) effects produced by developing a new compound with simultaneous agonistic activity at NOP receptor and MOP receptor.

**Figure 3 molecules-27-00595-f003:**
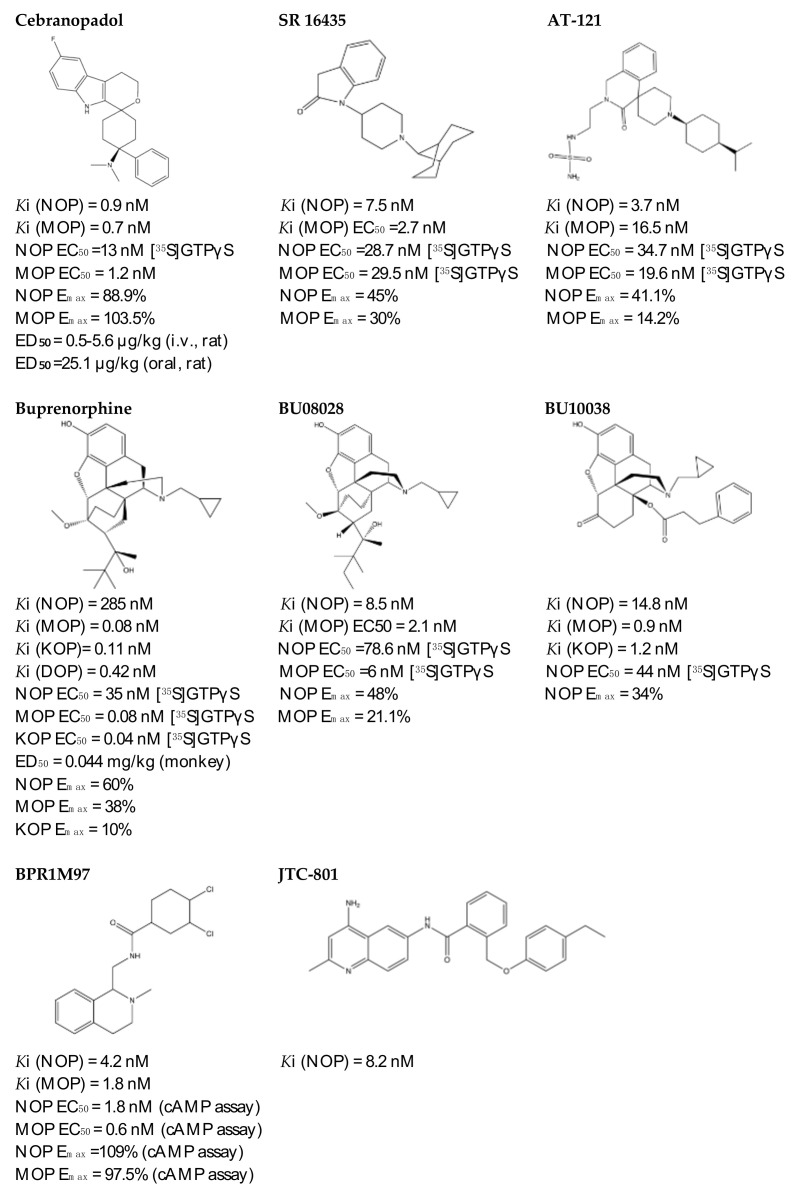
Chemical structures and in vitro pharmacological profiles of bifunctional and mixed NOP receptor ligands that target pain [99,110,111,112,113,114,115,116,117,118,119].

**Table 1 molecules-27-00595-t001:** The peptides that have antinociceptive activity are summarized.

Name/Structure	Category	In Vitro Human NOP Receptor			In Vivo		Ref
		Receptor Binding pKi	[^35^S]GTPγS pK_B_/pA_2_	Ca^+2^ Mobilization pK_B_/pA_2_	Administration Route/Dose/Species	Effect	
[Nphe^1^]N/OFQ(113)NH_2_	Selective NOP receptor antagonist	8.39	7.33	6.29	(30 nmol) i.c.v. mice	Analgesia Promote morphine-induced analgesia.	[61]
[Nphe^1^, Arg^14^, Lys^15^]N/OFQ-NH_2_ (UFP-101)	Selective NOP receptor antagonist	10.24	8.85	7.66	(10 nmol) i.c.v. mice	Long lasting analgesia Block N/OFQ effect on locomotor activity	[62]
					(10 nmol) i.t. mice	Block N/OFQ (i.t.1 nmol) analgesic effect	[66]
[(pF)Phe^4^Aib^7^Arg^14^Lys^15^]N/OFQ-NH_2_ (UFP-112)	Selective NOP receptor agonist	10.55	10.55	9.05	(1–100 pmol) i.c.v. mice	Hyperalgesia Decrease locomotor activity	[67]
					(1–100 pmol) i.t. mice	Long lasting dose dependent analgesia	[67]
					(0.1 and 10 nmol/kg) Intravenous (i.v.) rats	Decrease heart rate Decrease blood pressure Decrease urinary sodium excretion Increase urine flow	[67]
					(1–10 nmol) i.t. monkey	Dose-dependent analgesia without inducing itch Promotes morphine-induced analgesia without increasing itch response	[68]
[Phe^1Ψ^(CH_2_-NH)Gly^2^(pF)Phe^4^Aib^7^Arg^14^Lys^15^]N/OFQ-NH_2_ (UFP-113)	Selective NOP receptor partial agonist	10.26	9.72	7.97	(0.001–1 nmol) i.t. rats	Analgesia	[69]
PWT2-N/OFQ 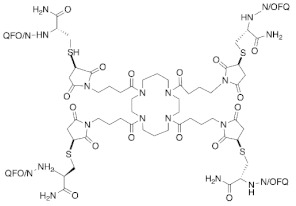	Selective NOP receptor agonist	10.3	10.12	8.83	(250 pmol) i.c.v. mice	Decease locomotor activity	[70]
					(2.5–250 pmol) i.t. mice	Dose-dependent analgesia	[71]
					(0.3, 1, and 3 nmol) i.t. monkey	Analgesia No itching No sedation No impairment in motor activity	[71]

**Table 2 molecules-27-00595-t002:** Non-peptide NOP receptor ligands targeting pain.

Name/Structure	Category	In Vitro Human NOP Receptor			In Vivo		Ref
		Receptor Binding pKi	[^35^S]GTPγS pEC_50_	Ca^+2^ MobilizationpEC _50_	Administration Route/Dose/Species	Effect	
Ro 65-6570 _ 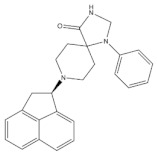 _	NOP receptor non peptide agonist	8.6			(0.1–1 mg/kg) (0.03 to 1 μmol/kg) i.v. mice	Analgesia	[79,80]
Ro 64-6198 _ 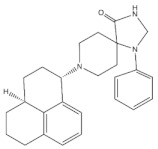 _	NOP receptor nonpeptide agonist	9.41	8.09	7.98	(3 mg/kg) (1 mg/kg) intraperitoneal (i.p) mice (0.3 to 3 mg/kg) i.p. mice	Analgesia Additive analgesia anxiolytic-like effects	[81,82,83]
					(0.001–0.06 mg/kg), subcutaneous (s.c.) monkey	Analgesia No depression No itching No reinforcing	[17]
SCH221510 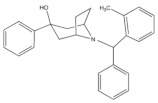	Selective NOP receptor nonpeptide agonist	0.3	12		1–30 mg/kg) peroral (p.o.) rat	anxiolytic-like effects	[84]
					(0.1–3.0 mg/kg) i.p., p.o., intracolonic mice	potent anti-inflammatory and analgesic effect	[85]

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
