# Peer review of "Spotlight on Nociceptin/Orphanin FQ Receptor in the Treatment of Pain"

_molecules, 2022, doi:10.3390/molecules27030595_

Round 1

Reviewer 1 Report

Dear Authors,

 In attachment there are a few suggestions, if you think that they can help you to improve your manuscript.

This review clearly and in detail describe structure, distribution, and signaling pathway of NOP receptor regarding to pain. Also, it is describe different NOP ligands as a potential alternative for conventional opioid analgesic. The significance of this review are to summarise available data and it helpful for rational design of new safer opioid analgesics.
The term opiate is often mentioned in the manuscript, I think the more appropriate word is opioid.
Given the potential therapeutic significance of cebranopadol, it is useful to highlight clinically available pharmacokinetic data and detailed safety data.

 Kind regards,

Author Response

Dear Authors,

In attachment there are a few suggestions, if you think that they can help you to improve your manuscript.

This review clearly and in detail describe structure, distribution, and signaling pathway of NOP receptor regarding to pain. Also, it is describe different NOP ligands as a potential alternative for conventional opioid analgesic. The significance of this review are to summarise available data and it helpful for rational design of new safer opioid analgesics. The term opiate is often mentioned in the manuscript, I think the more appropriate word is opioid. Given the potential therapeutic significance of cebranopadol, it is useful to highlight clinically available pharmacokinetic data and detailed safety data.

Response: As kindly suggested, the term opiate has been replaced by opioid in the manuscript. In addition, the cebranopadol’s pharmacokinetic profiles and detailed safety data have been added (page 13, lines 787-789 and 796-808).

Reviewer 2 Report

The review mostly focuses on the ligands of opioid peptide receptor nociceptin.

-Nociceptin is misspelled as nociception in the title and in the abstract.

-Abstract and introduction: I don't think the abstract and the introduction reflect the main focus of the review. The review mainly explains NOP ligands. Yet, in both introduction and abstract the emphasis is on the and side effects / misuse of opioid agonists. Both introduction and abstract should be rearranged focusing on NOP and NOP ligands.

-Structure of NOP: I think this section is too detailed. Could be shortened considering the main emphasis is NOP ligands.

Author Response

The review mostly focuses on the ligands of opioid peptide receptor nociceptin.

  1. Nociceptin is misspelled as nociception in the title and in the abstract.

Response: We thank the reviewer for pointing out the spelling error of a title. Nociception opioid receptor has been replaced by nociceptin/orphanin FQ receptor in the title and abstract.

  1. Abstract and introduction: I don't think the abstract and the introduction reflect the main focus of the review. The review mainly explains NOP ligands. Yet, in both introduction and abstract the emphasis is on the and side effects / misuse of opioid agonists. Both introduction and abstract should be rearranged focusing on NOP and NOP ligands.

Response: Both introduction and abstract have been rearranged as kindly suggested.

  1. Structure of NOP: I think this section is too detailed. Could be shortened considering the main emphasis is NOP ligands.

Response: Structure of NOP has been shortened as kindly advised.

Reviewer 3 Report

This review article described the nociceptin opioid receptor and its ligands from the viewpoint of pain treatment. In early days when nociceptin was discovered, the NOP receptor was skeptical as a drug target for analgesics because its antinomic pharmacological effects, antinociceptive and nociceptive effects. However, the NOP receptor is recently focused as promising target for analgesics because NOP agonists are reported to show antinociceptive effects in primates regardless of their administered doses and administration routes. Therefore, I think that this review deals with a timely subject and useful for many readers. However, the authors should address the following points before the publication.

  1. While NOP is an approved nomenclature, MOR, DOR, and KOR are non-approved ones (Table 1 in reference 26). The authors should use only approved nomenclatures or non-approved ones.

  1. It is unclear the criteria in choosing the compounds. Why is not the selective agonist SCH221510 dealt with? Are there no reports showing antinociceptive effects of the compound? How about bifunctional ligands BU08028 and BU00038? Some other review articles picked up these compounds. Antagonist JCT-801 has completed its Ph2 clinical trials for neuropathic and postoperative pain. However, there is no comment about the compound (see J. Med. Chem. 2016, 59, 7011(doi: 10.1021/acs.jmedchem.5b01499)). On the other hand, I think that buprenorphine is not usually classified in NOP agonist. Buprenorphine and its metabolite nor-buprenorphine have complicated pharmacological profiles: buprenorphine is potent MOP and KOP agonist with low efficacy, less potent NOP agonist with moderate efficacy, and not agonist for the DOP; nor-buprenorphine is potent MOP, KOP, and DOP agonist with moderate to high efficacy and full NOP agonist with very low potency (J. Pharmacol. Exp. Ther. 2001, 297, 688.).

  1. I appreciate the authors to provide their opinion in the conclusion section. Bifunctional ligands would be more promising. However, bifunctional ligands have complicated in vitro pharmacological profiles. I strongly recommend the authors to describe these ligands and their pharmacological effects in the section 4.3 with showing their in vitro profiles for opioid receptors (affinity, potency, and efficacy). Moreover, the strength of antinociceptive effects such as %MPE and ED50 values should be indicated. It is also important to discuss the side effects.

  1. The peptide sequences are shown for the peptide ligands, whereas the chemical names are provided for non-peptide compounds. Such manner is acceptable. However, I strongly recommend the authors to delineate the chemical structures of all non-peptide compounds, which would help the readers to understand the structures of non-peptide compounds.

  1. It seems that some recent reviews dealing with NOP ligands are not cited. For example, Neurosci. Rep. 2020(doi: 10.1002/jnr.24624), Curr. Med. Chem. 2018, 25, 2353(doi: 10.2174/0929867325666180111095458), J. Med. Chem. 2016, 59, 7011(doi: 10.1021/acs.jmedchem.5b01499).

  1. There are many typos and some inappropriate expressions. The authors must carefully review the manuscript again and make appropriate revisions. The followings are some examples.
  • In several parts including the title, “nociception opioid receptor” is used. The correct expression is “nociceptin opioid receptor.”
  • For some compounds’ names, brackets were absent.
  • Is “early systemic SAR” (line 217) correct? Systematic?
  • “Synthetic NOP agonist” is inappropriate for Ro64-6198 because both non-peptide and peptide compounds are obtained by syntheses.
  • What is GTPγ3S in Table? [35S]GTPγS?
  • In the chemical name, “R,” “S” (showing stereochemistry), and indicated “H” must be shown in italic face.
  • What is “pain selectivity”? (line 343)
  • “Using selective MOR and NOP receptor antagonists, J-113397 and naltrexone, respectively (lines 394-395)” is incorrect. J-113397 is NOP antagonist, while naltrexone is MOP antagonist when it is used in an appropriate dose (when naltrexone was administered at a high dose, it functions as the universal opioid antagonist).
  • What is “favorable side effects”? (line 430) I think that side effects are usually unfavorable.

Author Response

This review article described the nociceptin opioid receptor and its ligands from the viewpoint of pain treatment. In early days when nociceptin was discovered, the NOP receptor was skeptical as a drug target for analgesics because its antinomic pharmacological effects, antinociceptive and nociceptive effects. However, the NOP receptor is recently focused as promising target for analgesics because NOP agonists are reported to show antinociceptive effects in primates regardless of their administered doses and administration routes. Therefore, I think that this review deals with a timely subject and useful for many readers. However, the authors should address the following points before the publication.

  1. While NOP is an approved nomenclature, MOR, DOR, and KOR are non-approved ones (Table 1 in reference 26). The authors should use only approved nomenclatures or non-approved ones.

      Response: As kindly pointed out, the approved nomenclature, MOP, DOP, and KOP have been  

      used in the manuscript.

  1. It is unclear the criteria in choosing the compounds. Why is not the selective agonist SCH221510 dealt with? Are there no reports showing antinociceptive effects of the compound? How about bifunctional ligands BU08028 and BU00038? Some other review articles picked up these compounds. Antagonist JCT-801 has completed its Ph2 clinical trials for neuropathic and postoperative pain. However, there is no comment about the compound (see Med. Chem. 2016, 59, 7011(doi: 10.1021/acs.jmedchem.5b01499)). On the other hand, I think that buprenorphine is not usually classified in NOP agonist. Buprenorphine and its metabolite nor-buprenorphine have complicated pharmacological profiles: buprenorphine is potent MOP and KOP agonist with low efficacy, less potent NOP agonist with moderate efficacy, and not agonist for the DOP; nor-buprenorphine is potent MOP, KOP, and DOP agonist with moderate to high efficacy and full NOP agonist with very low potency (J. Pharmacol. Exp. Ther. 2001, 297, 688.).

Response: This insightful comment is well taken. SCH221510 (selective non-peptide NOP agonist), BU08028 and BU00038 (bifunctional ligands), and JCT-801 (NOP antagonist) have been described in the manuscript (page 10, lines 590-600; page 12, lines 731-739, page 12; lines 751-757, and pages 12-13, lines 758-773).

  1. I appreciate the authors to provide their opinion in the conclusion section. Bifunctional ligands would be more promising. However, bifunctional ligands have complicated in vitro pharmacological profiles. I strongly recommend the authors to describe these ligands and their pharmacological effects in the section 4.3 with showing their in vitro profiles for opioid receptors (affinity, potency, and efficacy). Moreover, the strength of antinociceptive effects such as %MPE and ED50 values should be indicated. It is also important to discuss the side effects.

Response: As kindly advised, the in vitro profiles for bifunctional ligands, the available data that is related to ED50 values and side effects have been added.

  1. The peptide sequences are shown for the peptide ligands, whereas the chemical names are provided for non-peptide compounds. Such manner is acceptable. However, I strongly recommend the authors to delineate the chemical structures of all non-peptide compounds, which would help the readers to understand the structures of non-peptide compounds.

Response: As kindly advised, the chemical structures of all non-peptide compounds have been provided in a new Figure (Figure 3).

  1. It seems that some recent reviews dealing with NOP ligands are not cited. For example,  Rep.2020(doi: 10.1002/jnr.24624), Curr. Med. Chem. 2018, 25, 2353(doi: 10.2174/0929867325666180111095458), J. Med. Chem. 2016, 59, 7011(doi: 10.1021/acs.jmedchem.5b01499).

      Response: As kindly advised, these three reviews have been cited (reference number 108,  

      109, and 120).

  1. There are many typos and some inappropriate expressions. The authors must carefully review the manuscript again and make appropriate revisions. The followings are some examples.
  • In several parts including the title, “nociceptionopioid receptor” is used. The correct expression is “nociceptin opioid receptor.”

Response: We thank the reviewer for pointing out the spelling error of a title. Nociception opioid receptor has been replaced by nociceptin/orphanin FQ receptor in the title and abstract.

  • For some compounds’ names, brackets were absent.

Response: We thank the reviewer for noticing that. All missing brackets that are related to peptide NOP ligands have been added.

  • Is “early systemic SAR” (line 217) correct? Systematic?

Response: We thank the reviewer for pointing out this error. Systemic SAR has been replaced by systematic (page 5, line 406).

  • “Synthetic NOP agonist” is inappropriate for Ro64-6198 because both non-peptide and peptide compounds are obtained by syntheses.

Response: As kindly instructed, the synthetic term has been deleted.

  • What is GTPγ3S in Table? [35S]GTPγS?

Response: [35S]GTPγS has been used instead of GTPγ3S in the manuscript including tables as kindly advised.

  • In the chemical name, “R,” “S” (showing stereochemistry), and indicated “H” must be shown in italic face.

Response: The italic face has been used to indicate the stereochemistry of NOP ligands.

  • What is “pain selectivity”? (line 343)

Response: We thank the reviewer for pointing out this error. The word selectivity has been replaced by sensitivity (page 10, line 576).

  • “Using selective MOR and NOP receptor antagonists, J-113397 and naltrexone, respectively (lines 394-395)” is incorrect. J-113397 is NOP antagonist, while naltrexone is MOP antagonist when it is used in an appropriate dose (when naltrexone was administered at a high dose, it functions as the universal opioid antagonist).

Response: This insightful comment is well taken. A MOP selective low dose of naltrexone was used in that experiment to specifically antagonize the MOP activity (page 12, lines 716-717).

  • What is “favorable side effects”? (line 430) I think that side effects are usually unfavorable.

Response: We thank the reviewer for pointing out this error. Limited range of unwanted effects has been used instead of favorable side effects (page 13, line 789).

Round 2

Reviewer 2 Report

I have no further comments.

Author Response

Thank you.

Reviewer 3 Report

The authors well addressed the reviewers” comments. Figure 3 in the revised manuscript is very informative. However, agonists have two indices, potency (EC50 value) and efficacy (Emax value). The Emax values of agonists should be also showed in Figure 3.

Author Response

Dear Prof. Dr. Mariana Spetea and Prof. Dr. Richard M. van Rijn,

We are submitting here with the revised version of our review entitled “Spotlight on nociceptin/orphanin FQ receptor in the treatment of pain”. Again, we wish to thank you and the reviewers for the constructive comments/suggestions on our manuscript. The modifications we have made in response to the issues raised are as follows.

Reviewer #3 (Round 2):

The authors well addressed the reviewers” comments. Figure 3 in the revised manuscript is very informative. However, agonists have two indices, potency (EC50 value) and efficacy (Emax value). The Emax values of agonists should be also showed in Figure 3.

Response: As kindly advised, the reported efficacy represented by Emax values for NOP ligands that target pain have been added to the revised Figure 3 (page 14).